# Patient perceptions by race of educational animations about living kidney donation made for a diverse population

Maria M. Keller [1,2]*, Todd Lucas[3,4], Renee Cadzow[5], Thomas Feeley[6], Laurene Tumiel Berhalter[7,8,9], Liise Kayler[2,9]

1 Department of Community Health and Behavior, University at Buffalo, State University of New York, Buffalo, New York, United States of America, 2 Transplant and Kidney Care Regional Center of Excellence, Erie County Medical Center, Buffalo, New York, United States of America, 3 Division of Public Health, College of Human Medicine, Michigan State University, Flint, Michigan, United States of America, 4 Department of Epidemiology and Biostatistics, College of Human Medicine, Michigan State University, East Lansing, Michigan, United States of America, 5 Department of Health Services Administration, D'Youville College, Buffalo, New York, United States of America, 6 Department of Communication, University at Buffalo, State University of New York, Buffalo, New York, United States of America, 7 Department of Family Medicine, University at Buffalo, State University of New York, Buffalo, New York, United States of America, 8 University at Buffalo, Clinical and Translational Science Institute, Buffalo, New York, United States of America, 9 Jacobs School of Medicine and Biomedical Sciences at the University at Buffalo, Buffalo, New York, United States of America

* mariakel@buffalo.edu

**Data Availability Statement:** Data cannot be shared publicly because the interview transcripts contain personal information that potentially identifies participants and would breach participant

## Abstract

### Introduction

This qualitative study sought to identify potential design and delivery alterations to inform cultural adaptation of educational animations about living donor kidney transplantation (LDKT)–previously developed for a diverse population–to better fit Black Americans' needs.

### Methods

We conducted a secondary analysis of 88 transcripts derived from interviews and focus groups conducted with diverse target users (62 kidney failure patients, 36 prior/potential donors, and 11 care partners) to develop 12 animations about LDKT, named *KidneyTIME*. Statements were abstracted and coded pertaining to cognitive and communication barriers to LDKT, and the perceived value of using the videos to learn and share the information with social network members using content analysis. Incidence counts of each content code were also calculated to assess differences between Black and non-Black patients.

### Results

Cognitive barrier codes included lack of knowledge, ambivalence, and concern for donor. Communication barrier codes included reluctance and difficulty talking about LDKT. Cognitive facilitating codes included attention-getting, efficient learning, manageable content, emotional impact, and new knowledge. Communication facilitating codes included delivery through many dissemination channels and broadly shareable. Compared to non-black

confidentiality if made publicly available. Release our data publicly is problematic since logical supposition of who the source is by members of the same community is possible and could jeopardize the anonymity of our participants and social network ties that arises from their shared history with others relating to kidney donation requests. In selecting particular quotes for the paper, we have ensured that no potentially identifiable information is included but this information is woven throughout the interview transcripts and any suitably redacted or anonymized transcripts would be so unserviceably thin that they would be devoid of meaningful content. We request that researchers interested in viewing these data contact Zachary Chakan, the University at Buffalo Internal Review Board Administrator who is assigned to this project: znchakan@buffalo.edu.

**Funding:** This study is supported by the Health Resources and Services Administration (HRSA) of the U.S. Department of Health and Human Services (HHS) as part of an award totaling $1.2M with no percentage financed with non-governmental sources. The contents are those of Dr. Liise Kayler and do not necessarily represent the official views of, nor an endorsement by, HRSA, HHS, or the U.S. Government. For more information, please visit HRSA.gov.

**Competing interests:** The authors have no relevant financial or non-financial interests to disclose.

patients (n = 33) Black patients (n = 29) more often stated concern for donor and reluctance/difficulty talking about LDKT as barriers, and less often stated efficient learning and manageable content as facilitators.

## Conclusion

Findings highlight the value of LDKT informational content that is visually appealing, digestible, non-threatening, and highly shareable. Heterogeneity may exist when considering access and intervention preferences in using *KidneyTIME* videos and highlight a potential for further cultural targeting or tailoring.

## Introduction

Well-designed educational resources that resonate with the intended users' questions and concerns are imperative for informed patient health decision-making. Animated video holds great promise for education and is increasingly being utilized by researchers [1]. We recently developed a single educational product of animated videos about living-donor kidney transplantation (LDKT) intended for a diverse audience [2]. To enhance the responsiveness of the animated videos to Black Americans, the predominant minoritized group in our community, we incorporated feedback from an independent subgroup of participants who self-identified as Black or African American during the development process [2]. Although crafted to enable acceptability of video materials to Black Americans, different message strategies and channel choices may be required to optimally influence different sub-groups. Health communication endeavors often provide opportunities to also develop and consider culturally targeted strategies. Animated video content in particular can be readily adapted to include culturally relevant adaptations, highlighting a great potential to refine educational products in specific cultural contexts using culturally responsive approaches. The decision on how to adapt materials for different audiences should be made in the context of the degree of actual heterogeneity with regard to influences on behavior, channel choices, and responsiveness to message execution [3].

A goal of our research is to reduce disparities of Black Americans to access LDKT, the best treatment for kidney failure. Black Americans are disproportionately burdened by kidney failure, but are far less likely than any other racial or ethnic groups to undergo LDKT [4]. Over the last two decades, the percentage of LDKTs has remained stable for non-Blacks but declined sharply for Black Americans by 43%, underscoring that those disparities are worsening [5]. Two critical hurdles that hinder all patients' journeys along the path to receiving a living-donor kidney transplant are misperceptions of LDKT and challenges asking for donation [6–13]. High quality education has been effective at increasing patient access to LDKT by changing the attitudes and behaviors of patients and their social network towards LDKT [14]. However, the effect of educational efforts has been less impactful among Black participants, potentially due to lower intervention uptake and fewer social network members reached [15].

Despite nearly two decades of educational development in kidney transplantation [4], few specific interventions have been designed to attenuate LDKT disparities, or to address the unique needs of Black Americans who face LDKT [16–19]. For example, prior educational interventions have been typically offered as in-person single sessions and may lack the flexibility necessary for participants who also face income or job-related restrictions and the associated time constraints or who live at distance, limiting the range of patients and their friends

and family who can be reached. This is concerning, as Black patients may need to reach out to a larger pool of potential living donors, beyond family members, since Black individuals who come forward as potential living donors are more likely to have disqualifying medical conditions (i.e., hypertension, diabetes) [20–23].

Since animated video education may offer individuals flexibility of learning and sharing information, we recently developed an educational animated video curriculum called *KidneyTIME* (Kidney donation and Transplant Information Made Easy) [2]. The *KidneyTIME* curriculum consists of a series of short, 2-dimensional animated videos optimized for release on small devices and social media channels to extend the reach of information. The videos were informed by health communication and multimedia best practices and iteratively developed with feedback from a diverse group of kidney failure patients and their social network members. A pre-post study showed the videos were promising to increase knowledge across groups of different races [24]. However, the extent to which the video education responds to LDKT cognitive and communication barriers reported specifically by Black individuals is unknown. Simultaneously, little is yet known about the potential to further the effectiveness of *KidneyTIME* through culturally targeted adaptations that can be readily implemented.

Therefore, the current study aims to present findings to consider general receptivity, evaluate potential differences in response to *KidneyTIME* among Black Americans, and also to consider the potential to develop culturally targeted *KidneyTIME* content to overlay general video educational materials for Black American viewers. We performed a secondary review of the original transcripts of interviews and focus groups that informed the *KidneyTIME* video design, including both formative and video development sessions. We extracted participants' statements pertaining to cognitive and communication barriers to access LDKT, as well as perceived cognitive and communication facilitation provided by the videos towards learning and sharing information about LDKT. We identified content codes for each statement and described participants' experiences and perceptions. We also quantified content code frequencies in each participant data in order to compare differences between Black and non-Black groups in response to *KidneyTIME*.

## Methods

A descriptive study of the original transcripts of interviews and focus groups used to inform the design of 12 videos about living donor kidney transplantation in the *KidneyTIME* curriculum was conducted to broadly inform potential changes to the content and dissemination of the videos to better fit Black Americans' needs. This study was approved by the University at Buffalo Institutional Review Board and follows the Consolidated Criteria for Reporting Qualitative Research (COREQ) checklist.

### Setting

Since 2018, a multidisciplinary team of transplant clinicians, researchers, and kidney disease stakeholders has been developing educational animations, called *KidneyTIME* [2], to improve access to kidney transplant information for all kidney failure patients and their social network in Buffalo, NY. Twelve videos about living donor kidney transplantation were completed in the Fall of 2019. The original research involved (1) formative discussions among kidney transplant candidates and recipients (hereafter referred to as patients) about their personal perspectives and experiences discussing live kidney donation with their friends and family and (2) video development cognitive interviews and focus groups with an expanded group of intended users, including patients' care partners and previous/potential living kidney donors (hereafter referred to as donors), to obtain feedback on video prototypes [2]. The interviews and focus

groups for the original study were conducted between 9/10/18-10/22/19 at the local transplant hospital where patients received their medical care.

## Sample

Recruitment procedures have been detailed elsewhere [2]. Briefly, trained research staff recruited kidney patients and donors associated with a local transplant program who were at least 18 years of age and English-speaking. Care partners were invited by participants. Research staff did not know potential participants before approaching them about study participation; they were approached in their clinic rooms and via opt-out letters and telephone calls to facilitate 50% Black patient enrollment.

## Information collection

Researchers obtained written informed consent by all participants and captured information on each participant's sociodemographic characteristics via a questionnaire at the start of each interview or focus group. Data collection was supervised by the study co-investigator, a qualitative methods expert (TF), and was conducted by any two of the following researchers trained in the conduct of cognitive interviews: a transplant surgeon known to the patients (LK), a research coordinator (DW, MS), a doctoral candidate in epidemiology (RS), or a doctoral candidate in behavioural science (MK). All interviewers were female. One was Black and the others were non-Hispanic White.

All sessions were conducted separately for donors and for Black patients (as determined by patient's self-reported race as being Black or African American) to increase participant comfort in discussing their experiences and allowing for separation of data to assess donor and Black patient perspectives.

The discussions used a semi-structured interview guide developed by the research team, which reflected the aims of the *KidneyTIME* video development research. To explore patients' personal perspectives and experiences of discussing live kidney donation with social network members, the questions were: (1) When you decided to get a kidney transplant, what did you think of getting one from a living donor? and (2) How did you go about finding a living donor? What response did you get? To inform video development, participants were shown 4–12 videos, with iterative changes made to reflect feedback from earlier participants. After viewing each video, participants were asked: (1) What are your thoughts about the video? (2) What do you like/dislike about the information provided in the video? (3) Was there any information that wasn't clear? Easy to understand? (4) What do you think about the look of the video? The graphics? Characters? To explore anticipated video sharing, participants were asked at the end of the session: Would you have shared the video(s) with your family and friends? How would you share the video(s)? Probes related to the above questions were included in the analysis. Other questions from the discussion phase are not relevant here and have been examined elsewhere [2].

All discussions were audio recorded and did not exceed 90 minutes. Participants received a $25 check to compensate for their time. Audio recordings were transcribed verbatim by a professional transcription company and de-identified to maintain confidentiality.

## Data analysis

The interview and focus group data were analyzed using a directed content analysis approach [25] to focus the data on four *a priori* topics elicited by the interview questions: (1) cognitive barriers to LDKT, (2) communication barriers to LDKT, (3) the perceived value of using *KidneyTIME* videos to learn about LDKT, and (4) the anticipated sharing of the videos with social

network members. To begin, a researcher (MK) reviewed the transcripts in their entirety, listened to recordings in order to conceptualize the data, and highlighted statements that appeared to describe each topic. All highlighted text was then independently coded within the transcripts by MK and LK, identifying further statements as necessary. Coders compared, discussed, and revised the codes until consensus was reached. The coded segments corresponding to each topic were imported into a spreadsheet (Excel, Microsoft Corp, WA, US) and linked to each participant to facilitate quantitative analysis. The incidence of each code category was counted for each participant and compared by racial group using percentages (i.e., the number of participants expressing at least one statement pertaining to the code category divided by the total number of participants within the race group). A meaningful difference between groups was considered 10%, large enough to be of practical significance. At the beginning of the analyses, statements from patients, care partners, and donors were coded and examined separately. However, it became clear that each group's results were closely related to those of the other. The researchers decided to merge the statements during the final description of the content code categories. Along with the 2 researchers, a senior researcher (TF) accompanied the entire process and was actively involved in numerous discussions concerning topics and codes. Last, MK selected representative quotations to illustrate code content (Table 3). Study participants were not involved in the analysis process or in confirming the accuracy of the transcripts and findings.

## Results

There was a total of 88 transcripts of interviews and focus groups with 62 patients, 36 donors, and 11 care partners (76 interviews, 12 focus groups with 2 to 5 participants) (Table 1). Of the 12 focus groups, 3 were conducted as homogeneous donor groups and 4 as homogenous patient groups, while the remaining 5 mingled patients with care partners. Among patients, 29 were Black, and 33 were non-Black (predominantly non-Hispanic White). Among care partners/donors, 6 were Black, and 41 were non-Black. Participants ranged in age between 24 and 81 years, with a mean age of 52 years. The majority were female (61%) and married or living

**Table 1. Participant characteristics.**

| Sample | Characteristic Mean (range) or n (percent) | Black participants | Non-Black participants |
|---|---|---|---|
| | | **(n = 29)** | **(n = 33)** |
| Transplant candidates and recipients | Age (years) | 55 (28–76) | 57 (29–81) |
| | Sex–Male | 13 (45%) | 12 (38%) |
| | Education–Less than college | 20 (68%) | 11 (32%) |
| | Married/Living with someone | 10 (34%) | 24 (72%) |
| | Income < $30,000 | 10 (34%) | 1 (3%) |
| | Transplant candidates | 7 (24%) | 3 (9%) |
| | Transplant recipients | 22 (76%) | 30 (91%) |
| Donors and care partners | | **(n = 6)** | **(n = 41)** |
| | Age (years) | 58 (47–68) | 47 (24–78) |
| | Sex–Male | 3 (50%) | 12 (29%) |
| | Education–Less than college | 2 (33%) | 6 (15%) |
| | Married/Living with someone | 2 (33%) | 31 (76%) |
| | Income < $30,000 | 5 (83%) | 6 (15%) |
| | Prior or potential donor | 0 (0%) | 36 (88%) |
| | Care partner | 6 (100%) | 5 (12%) |

with someone (62%). Black and non-Black participants differed in terms of educational achievement, with fewer Black individuals attending college. Black individuals also reported lower household incomes, with over half earning less than $30,000 annually. Detailed characteristics are reported in Table 1.

## Content categories

Based on secondary review of 88 transcripts, we identified participant statements covering the 4 topics determined *a priori*. On the topic of cognitive barriers of patients about LDKT, there were 3 content categories: ambivalence, lack of knowledge, and concern for donor. On the topic of communication barriers of patients about LDKT, there were 2 content categories: reluctance to talk and difficulty talking about LDKT. On the topic of video as cognitive facilitators among all participants, there were 5 content categories: attention-getting, efficient learning, manageable content, emotional impact and new knowledge. On the topic of video as communication facilitator among all participants, there were 2 content categories: many dissemination channels and broadly shareable. Within the content categories, we describe experiences and perceptions of patients, donors, and care partners, and quantitatively compare patient-only responses by race group. An overview of the quantitative findings is provided in Table 2. Illustrative quotes are provided in Table 3.

## Cognitive barriers to LDKT

Content categories that emerged as cognitive barriers to access LDKT were: ambivalence, lack of knowledge, and concern for donor.

*Ambivalence*. Ambivalence about LDKT was derived from patients feeling a lack of urgency for a transplant. Patients not yet requiring dialysis felt they had time to find a donor. Those already on dialysis thought that dialysis would keep them alive until a deceased-donor kidney

**Table 2. Number and proportion of Black and non-Black patient-participants that made at least one statement pertaining to each content code category.**

| Content Code Category N (%)[1] | Black patients (N = 29) | Non-Black patients (N = 33) | Delta* |
|---|---|---|---|
| **Cognitive Barriers** | | | |
| 1. Ambivalence | 1 (3%) | 4 (12%) | 9% |
| 2. Lack of knowledge | 2 (7%) | 4 (12%) | 5% |
| 3. Concern for the donor | 12 (41%) | 1 (3%) | **38%** |
| **Communication Barriers** | | | |
| 1. Reluctance to talk about LDKT | 7 (24%) | 4 (12%) | **12%** |
| 2. Difficulty talking about LDKT | 8 (28%) | 5 (15%) | **13%** |
| *KidneyTIME* **Videos as Cognitive Facilitators** | | | |
| 1. Attention-getting | 8 (28%) | 11 (33%) | 5% |
| 2. Efficient learning | 2 (7%) | 9 (27%) | **20%** |
| 3. Manageable content | 3 (10%) | 9 (27%) | **17%** |
| 4. Positive impact | 7 (24%) | 6 (18%) | 6% |
| 5. New knowledge | 7 (24%) | 8 (24%) | 0% |
| *KidneyTIME* **Videos as Communication Facilitators** | | | |
| 1. Many dissemination channels | 12 (41%) | 18 (55%) | 6% |
| 2. Broadly shareable | 7 (24%) | 10 (30%) | 6% |

[1]Number of participants expressing at least one statement within the code category divided by the total number of participants within the race group.

*Between-group differences of 10% or greater are emboldened.

**Table 3. Illustrative quotes for content categories within each topic.**

| Content Category | Illustrative quotes |
| --- | --- |
| **Topic: Cognitive Barriers** | |
| Ambivalence | • Maybe if I was on dialysis, I would be asking more people, but you know, I wasn't so, you know. (Non-Black, post-transplant, 46–50, female, high school/trade school, interview)<br>• I felt like I caused my own demise. . .Therefore, I'm going to wait it out. (Black, post-transplant, 26–30, male, college, focus group)<br>• I had just accepted I was gonna die. I didn't wanna put that on my [parent]. (Black, post-transplant, 26–30, female, less than high school, interview) |
| Lack of knowledge | • I think in my belief that there really wasn't a difference [between living donors and deceased donors] because, I mean, obviously if they are giving it to you, it was a still functioning kidney. (Black, post-transplant, 66–70, female, less than high school, interview)<br>• Yes, I would like to have [a living donor kidney], but at the time it was such a long waiting period. (Non-Black, listed, 46–50, female, college, interview)<br>• The doctors I had back then. . .they were telling me that I could not accept a living donor kidney from a relative, so I had to get a cadaver donor. . .They didn't tell me I could get a living related donor. (Black, post-transplant, 46–50, female, high school/trade school, interview) |
| Concern for donor | • Yeah [they offered] and then I said no because you know what if something's wrong with the other one then you have to go to dialysis. I wouldn't take it. (Black, post-transplant, 36–40, female, college, interview)<br>• I didn't want her to give because she was so young, you know, and I didn't want to, um, you know, stop her from doing everything. (Black, post-transplant, 51–55, male, high school/trade school, interview)<br>• I was concerned about her having another baby. (Black, post-transplant, 51–55, male, high school/trade school, interview)<br>• What I was told . . .they go through more tests than you do. . .I would never want to put somebody through that. (Black, post-transplant, 66–70, female, less than high school, interview)<br>• How much pain is he gonna be in? Because I didn't want him in a lot of pain. (Black, post-transplant, 66–70, female, less than high school, interview)<br>• That's when it really hit me, this isn't just [my donor] doing this for me, his whole family is gonna be affected by this. And that was the first time it kinda hit and I was like, oh God, don't let anything happen to [my donor]. (Black, post-transplant, 66–70, female, less than high school, interview)<br>• I just wouldn't be comfortable asking them to take that. . .as far as recovery, the pills. (Black, post-transplant, 56–60, female, college, interview)<br>• [My siblings] offered me a kidney, and I told them no. I didn't want to put anyone out of work or tag them down. (Non-Black, post-transplant, 61–65, female, less than high school, interview) |
| **Topic: Communication Barriers** | |
| Reluctance to talk about LDKT | • It's hard for me to ask people for things. (Non-Black, listed, 66–70, female, college, focus group)<br>• I don't want to ask people for help in general. I mean that's a huge thing to ask someone, and I didn't want them to feel like obligated in any way that they had to do it. (Non-Black, listed, 31–35, female, college, interview)<br>• Because I'm from a very large family, and you would think everybody knows I need a kidney, somebody would step forward. . .No one stepped up, and I don't ask; they all know. (Black, post-transplant, 46–50, female, less than high school, interview)<br>• So, you are kind of asking but not asking [by showing the videos]. (Non-Black, post-transplant, 31–35, male, college, interview)<br>• I wouldn't even want to ask them because I don't want them to say no. I already know they would say it, I don't want to hear it. (Black, post-transplant, 56–60, female, high school/trade school, interview)<br>• My reasoning was more like, I don't want to put it on people just right there (Black, post-transplant, 51–55, male, high school/trade school, interview)<br>• We were afraid because kidney failure it runs, it runs in our family. (Black, 36–40, post-transplant, female, college, interview)<br>• I didn't want anybody asking about it or anything like that because I feel like that's none of their business. (Black, post-transplant, 56–60, female, high school/trade school, interview) |
| Difficulty talking about LDKT | • I wouldn't know how to begin to walk up to someone and ask them something like that. (Black, post-transplant, 56–60, female, college, interview)<br>• How do you even say it? How do you ask somebody? (Non-Black, post-transplant, 56–60, male, college, focus group)<br>• With my family, I know that to ask them to even get tested would be so hard. (Black, post-transplant, 56–60, female, high school/trade school, interview)<br>• I didn't want them to feel, like, obligated in any way that they had to do it (Non-Black, post-transplant, 31–35, female, college, interview)<br>• I asked a few people and they say they came and got tested but I don't know if they did or didn't. (Black, post-transplant, 51–55, male, less than high school, interview)<br>• A couple of people have a, like my relatives have said, you know, I called and, you know, I say, 'thank you so much,' and um, I sort of leave it alone. I don't want to pester them or anything. (White, post-transplant, 31–35, college, interview)<br>• I had so many people offer to get tested and they just never called. . .I am no longer friends with any of those people because they literally just stopped talking to me after they realized they didn't want to do it. (Black, post-transplant, 66–70, female, high school/trade school, interview) |
| **Topic: *KidneyTIME* Videos as Cognitive Facilitators** | |

(*Continued*)

**Table 3.** (Continued)

| Content Category | Illustrative quotes |
|---|---|
| **Topic: Cognitive Barriers** | |
| Attention-getting | • Make the background color [black and white] a little less stark. . . make it a little bit more homey. Or even add, like, the sun, a window, flowers, something like that. (non-Black, post-transplant, 71–75, female, high school/trade school, interview)<br>• You see the visual and you know, so it's yeah, less boring. Something, something, uh, to keep your attention. (Black, post-transplant, 46–50, male, high school/trade school, interview)<br>• Cartoons are OK because it still caught my attention, whereas doctors are more monotone. (Non-Black, post-transplant, 46–50, female, college, interview)<br>• It's like. . .it's almost touchy-feely. If I can touch and feel it and see it. . . It's better than me lookin' at words on a piece of paper. (Black, post-transplant, 66–70, female, high school/trade school, interview) |
| Efficient learning | • Short and sweet, and right to the point. (Non-Black, post-transplant, 66–70, female, high school/trade school, focus group)<br>• They're simple enough that if you're just sitting there for a few minutes, you can take [the videos] in. (Non-Black, donor, 46–50, female, college, interview)<br>• Easy to follow, like not super formal. I don't feel like I have to pay attention super carefully. (Non-Black, donor, 21–25, female, college, interview)<br>• The book that they gave is a lot of statistics, and percentages, and numbers. It's a little more daunting to take in than [the videos]. (Non-Black, listed, 46–50, male, college, interview) |
| Manageable content | • They're nice, short little videos. It's not something that's super long in length because that deters me right away. (Non-Black, donor, 46–50, female, college, interview)<br>• You can put that in a video. It's a lot quicker to talk than it is to read and it would have been so much better. (Black, post-transplant, 56–60, female, high school/trade school, interview)<br>• If I had been able to watch a video, that would have been so much better than they give you, like, this giant folder with a million pages in it. (Black, post-transplant, 56–60, female, high school/trade school, interview)<br>• They're quick. They're not a 5-page research paper. (Non-Black, post-transplant, 51–55, female, college, interview) |
| Emotional impact | • I thought it was very non-threatening because the images were kind of cartoonish-like. You know, it wasn't, like, stern and it was very easy to look at it because it was just comfortable. (Black, post-transplant, 66–70, female, college, interview)<br>• I can imagine myself kind of plugging in to all that. (Non-Black, donor, 61–65, female, college, interview)<br>• I noticed her foot was stretched out like that and to me in represented the fact that you can walk now. You don't have to stop and pant and wait and you can just walk. It was wonderful. (Black, post-transplant, 36–40, female, college, interview) |
| New knowledge | • I just think that many people would think of themselves as not being able to. And that is like gentle and friendly and informative. (Black, post-transplant, 61–65, female, high school/trade school, interview)<br>• It would ease the barrier between people thinking they can't live if they donate 'cause of their financial situation. (Black, post-transplant, 66–70, female, high school/trade school, interview)<br>• I didn't realize that the um, the recipient's insurance pay for it and you know, that would be another thing that would have stopped me for asking. (Non-Black, post-transplant, 51–55, female, college, interview)<br>• If you don't have this information, you're thinking that by taking a kidney from somebody else, you're putting them through the same thing. . .but this information will let them know like oh so I could ask someone and it might not be that bad for them. They can go on and lead a happy life. Maybe I could ask. (Black, post-transplant, 56–60, female, college, interview)<br>• I think that one gives you more security about donating cause if there's a problem, they can correct it. Um, it makes you more confident and donate. (Non-Black, donor, 31–35, female, college, interview)<br>• Knowing that it's. . .it's not traumatic. . .it would ease their nerves. (Black, post-transplant, 66–70, female, high school/trade school, interview)<br>• There's a lot of fear out there. It's a lack of knowledge. (Non-Black, post-transplant, 61–65, female, college, interview)<br>• I think it would be a tool to explain a little bit more to maybe prevent some anxiety. (Non-Black, donor, 36–40, female, college, interview) |
| **Topic: *KidneyTIME* Videos as communication facilitators** | |
| Many dissemination channels | • [I would share with] the people closest to us. (Black, post-transplant, 51–55, male, college, interview)<br>• If I had a potential donor who was serious, I would sit down and show them in a heartbeat. (Non-Black, listed, 61–65, female, college, interview)<br>• [Show the videos] like in a group, so you know what, even though I don't have questions, somebody would bring up a question. (Non-Black, donor, 36–40, female, college, focus group)<br>• We would have probably had a viewing party and honestly made fun a little bit. . .because we're a little bit weird. (Non-Black, donor, 36–40, female, college, interview)<br>• I would show anybody who would want to get a kidney transplant or done had a kidney transplant. I would say, won't y'all come in and watch these videos with me. (Black, post-transplant, 51–55, male, less than high school, interview)<br>• I would use the videos to kind of bridge that conversation, instead of can I have your kidney. Like, here's a link, maybe watch these videos, and then have them come up with the idea. (Non-Black, post-transplant, 46–50, female, college, interview)<br>• Would you just take a look at these? If you don't want to watch it, you don't have to watch it, but I'm going to watch it. . . You know, so maybe they would be interested if I watch it. You can sit down and watch it too. (Black, post-transplant, 71–75, male, high school/trade school, interview)<br>• It would answer a whole lot of questions that I wouldn't have to answer. (Non-Black, post-transplant, 61–65, female, college, interview)<br>• It's sort of like the hook used to bring you in. (Non-Black, post-transplant, 51–55, female, college, interview)<br>• Show them that it's not as scary as it sounds. (Black, post-transplant, 46–50, male, less than high school, interview)<br>• I think people like watching these things online and would when they normally wouldn't like read a post. (Non-Black, donor, 31–35, female, college, interview)<br>• The more places that you get them out there, the more donors you're gonna get, 'cause people are afraid. (Non-Black, donor, 46–50, female, college, interview) |

(*Continued*)

**Table 3.** (Continued)

| Content Category | Illustrative quotes |
|---|---|
| **Topic: Cognitive Barriers** | |
| Broadly shareable | • If you had a family and you were trying to expose your children to something, and they don't need that much information, yep, they're. . .we collaboratively. . .it was a family decision. We were all interactive with the whole thing. To expose them to something like that, gives them some understanding. (Non-black, post-transplant, 61–65, female, college, focus group)
• My [spouse] tends to, the biggest pain to him, sometimes he needs to hear it from an expert. . . I always try and find someone else put together and then he believes it. (Non-Black, post-transplant, 66–70, female, college, interview)
• To expose them [the kids] to something like this gives them some understanding. (Non-Black, post-transplant, 61–65, female, college, focus group)
• I think this would have been much more simplified and easier for [my parents] to understand everything. (Non-Black, post-transplant, 61–65, female, less than high school, interview)
• I would [share the videos] because your family has to learn about all of this as well because they the ones that have to help you out as much as they can. (Black, post-transplant, 46–50, male, less than high school, interview)
• That you have comprehensive information on kidney transplant donors and how people can get involved even if they're not actually donating a kidney, how they can help by other actions and resources, getting involved with other resources to help the patient through the process. (Native American, post-transplant, 51–55, male, college, interview) |

became available. Ambivalence also stemmed from patients feeling undeserving of donation due to guilt over poor self-care causing their kidney failure or from feeling resigned to their health status, accepting that they may not get a kidney transplant and might die: "I had accepted I was gonna die. I didn't wanna put that on my [parent]."

*Lack of knowledge*. Unawareness of the advantages of a living-donor kidney, such as the higher quality and the shorter wait time, resulted in patients opting for a deceased-donor kidney instead. Patients' misunderstanding about their eligibility to receive a living-donor kidney transplant resulted in some believing that waiting for a deceased-donor was their only transplant option: "The doctors were telling me that I could not accept a living donor kidney from a relative so I had to get a cadaver donor."

*Concern for donor*. Concern for a potential donor's health and well-being was multifactorial. Patients' most frequent concern was that the donor might develop kidney failure and the unbearable guilt they would feel if they had taken a kidney from them: "What if something's wrong with the other one then you have to go to dialysis. I wouldn't take it." Other concerns were about the ability of a donor to fully function with a single kidney in regard to playing sports, going to school, and having long lives that might include marriage and having children: "I didn't want to, um, you know, stop her from doing everything." Patients did not want to subject others to the extensive testing to determine their eligibility to donate, which was perceived as more rigorous than the recipient process or would be an obstacle to a donor volunteering. Some concerns for the donor were more proximal to the surgery, such as the donor's pain after surgery and potential surgical complications. Complications were further worrisome for their impact on donor families. Lastly, patients did not want their loved ones to have to endure the recovery process or to affect the donor's employment: "I didn't want to put anyone out of work or tag them down."

Black (vs non-Black) participants were more likely to report the cognitive barrier of concern for donor health (41% vs. 3%). LDKT ambivalence and lack of knowledge were consistently reported between groups (Table 2).

## Communication Barriers

Content categories that emerged as communication barriers were reluctance to talk and difficulty talking about LDKT.

*Reluctance to talk about LDKT*. Patients were reluctant to talk about LDKT for many reasons. They did not want to ask for help in general and especially not for such a large gift as a kidney donation. Patients avoided asking for donation if they sensed that others did not want

to be approached or if they anticipated being rejected by a possible donor: "I don't want them to say no. I already know they would say it. I don't want to hear it." One patient would not start a conversation about donation due to misinterpretation of common approach advice from transplant professionals to express your need rather than directly "ask" for a kidney. While silently waiting, some patients hoped for unsolicited offers of donation, expecting that family and friends who knew of their need and had an interest in donating would offer a kidney without a request being made: "No one stepped up, and I don't ask; they all know." Others accepted that they would not have a living kidney donor and did not anticipate offers of donation, often due to a lack of healthy potential donors available, citing extensive family histories of kidney failure and comorbidities, such as diabetes, high blood pressure, and heart disease, which they understood precluded donation. Similarly, lifestyle choices, such as diet or alcohol use, were common reasons for not approaching a potential donor. Lastly, some were silent out of a desire to maintain privacy about their kidney disease.

*Difficulty talking about LDKT*. Patients' expressed difficulty talking about LDKT primarily from not having the right words, trying to avoid making others uncomfortable, and difficulty coping with rejection. Patients described not knowing how to approach a potential donor, such as how to introduce the topic of donation into conversations: "How do you even say it? How do you ask somebody?" Some worried about making others feel uncomfortable or placing them in a difficult position to respond or deny the request: "I didn't want them to feel, like, obligated in any way that they had to do it." Feeling unprepared to answer questions was an obstacle for a few, but most did not see it as their role to teach others about the donation process, indicating that they would refer questions to be answered by transplant center staff. Dealing with a rejection response was difficult for patients and threatened relationships. Indirect rejection came in the form of non-communication. Some patients described making follow-up inquiries about the potential donor's status but felt they were being annoying, causing them to give up their search altogether. Others interpreted the non-communication as loss of friendship: "I'm no longer friends with those people because they literally just stopped talking to me after they realized they didn't want to do it."

Black (vs non-Black) participants were more likely to report the communication barrier of reluctance to talk about LDKT (24% vs. 12%) and difficulty talking about LDKT (28% vs. 15%) (Table 2).

### *KidneyTIME* videos as cognitive facilitators

Content categories that emerged for the *KidneyTIME* videos as a facilitator of learning about LDKT were through: attention-getting, efficient learning, manageable content, emotional impact and new knowledge.

*Attention-getting*. The videos' attractiveness held learners' attention. Participants expressed that the animations were attractive, describing them as "eye-catching," "colorful," and "not drab," which made it easier for them to pay attention. Color was an important aspect. When color was absent in some of the early black and white video prototypes, participants consistently expressed dissatisfaction, calling them "boring," "mundane," and "stark," and participants requested the addition of color to make it "interesting to watch." When in-color versions were presented, participants' impressions were positive, and they reported that the videos were preferable to other learning modalities that they had previously experienced, such as a "less interesting" group education, "monotone" doctors, "boring" websites, and "strait-laced" PowerPoints.

*Efficient learning*. The video information was described as "simple" and "right to the point." The simplicity made the information efficient, "easy to take in" quickly without having to concentrate too much or to go back and view it again. In addition to explaining concepts simply, the concepts themselves were considered straightforward: "basic," "clean," and "without

distractions." Straightforward information was considered information that they needed to know and that was not too complex. The straightforwardness of the video information was considered preferable to previous experiences of receiving complex information, including explanations from providers and reading materials that were "daunting to take in" because of being laden with numbers and statistics.

*Manageable content.* The short, chunked format made the video education manageable. Participants described the videos as "short and sweet" and that they "explained enough in a good amount of time" and were presented in a "neater package." Information presented in short chunks translated to participants' feeling that they could easily learn "all" the education in "digestible steps." Participants also commented that the videos covered the same information as the reading materials they were given, but with the videos they didn't feel "bombarded" or "bored" by the education. Others mentioned that feeling emotional about their medical situation made it difficult to read "10 pages' worth of information." In general, the videos were preferable to the reading materials they had been given, which were described in unmanageable terms—"a giant folder with a million pages in it" or a "5-page research paper." Some preferred the video explanations over hearing from multiple providers, which was described as "overwhelming."

*Emotional impact.* The video characters were relatable and scenes had an emotional impact. Participants described their responses to the characters and to specific scenes in the videos. The characters were described as familiar, "comfortable," "non-threatening," and drawing their attention. Participants imagined themselves "plugging in" to the characters' journey and recalled emotions they felt while viewing specific scenes. A scene of the donor character recovering from surgery engendered a "wonderful" feeling according to a transplant candidate viewer. The same scene put a potential donor viewer "at ease." A scene of the donor and recipient characters growing old together in the video reinforced the concept of longevity after kidney donation and it reduced a patient's feeling of fear and guilt of accepting a kidney from her potential donor. In contrast, a patient recalled a scene of a donor character frowning in response to the possibility of kidney failure and reported feeling "negativity." Another felt "nervous" while viewing the donor operation shown in the video "because they do the clamping and the part that makes me nervous about it."

*New knowledge.* Declarations of new knowledge were made by participants while watching the videos. Preconceived notions of donor ineligibility were clarified. Myths about how donation is paid for were dispelled. New knowledge aligned with statements of reduced concerns. More comfort about the recovery period was reported. Financial concerns were eased. As a result of new knowledge, patients reported feeling better about asking others to donate and accepting a donor's offer. Patients opined that sharing the videos with potential donors would reduce the donor's anxiety and increase their confidence about donating.

Fewer Black (vs. non-Black) participants reported efficient learning (7% vs. 27%) and manageable content (10% vs. 27%) after watching videos. The codes of attention-getting, emotional impact, and new knowledge were similarly reported between race groups (Table 2). In this section, we included both positive and negative statements. So we coded what was important, not necessarily what existed. (Table 2). For example, the code "attention-getting" applied to statements recommending the addition of color to early video prototypes and to statements describing the visual appeal of color in later video prototypes.

### *KidneyTIME* videos as a communication facilitator

Two content categories emerged for the *KidneyTIME* videos as a communication facilitator by being: 1) deliverable through many dissemination channels and 2) broadly shareable.

*Deliverable through many dissemination channels*. Participants indicated that the videos could be disseminated through many channels, increasing their communication ease about LDKT. Participants suggested that the *KidneyTIME* videos could be shown in-person to initiate and inform conversations about LDKT, usually within small groups comprised of family members, "people closest to us," and individuals who were "serious" or "thinking" about donation, or in a larger group such as a "viewing party." Integrating the videos into conversations would prompt discussion and generate questions. Participants also anticipated sending the videos electronically. Electronic video sharing to specific individuals was anticipated to be via email or by recommending that friends or family view the videos online (on YouTube or a website). Electronic sharing was also anticipated to large groups via social media, Facebook most commonly. Sharing asynchronously was seen as an easier way for a patient to engage potential donors rather than making a verbal request and easier for the potential donor by providing them with information and allowing them to come to their own decision, instead of being put on the spot to answer a donation request. Other benefits of video sharing were to educate others, by answering "a whole lot of questions that I wouldn't have to answer," and to reduce anxiety—"show them that it's not as scary as it sounds." The opportunity of incorporating videos in social media posts was described as "a hook used to bring you in" and more engaging, since "people wouldn't normally read a post." One patient summarized the importance of widespread dissemination: "The more places that you get them out there, the more donors you're gonna get, cuz people are afraid."

*Broadly shareable*. Participants imagined sharing the videos with a wide range of social network members including partners, siblings, children, and grandchildren to improve their understanding and support of kidney transplantation and donation. Some imagined the videos being useful to inform others how to be an advocate for living kidney donation on behalf of the patient and to promote caregiving after kidney transplantation. Sharing the videos was perceived as more effective than explaining themselves, especially for children, older people, or doubters.

The content categories of videos facilitating communication ease by being deliverable through many dissemination channels and by being broadly shareable was consistent across groups (Table 2). In addition, 62% of Black and 55% of non-Black participants either described how they would share the videos or stated "yes" when asked about anticipated sharing.

## Discussion

In this study, we performed a secondary analysis of transcripts from *KidneyTIME* formative and development interviews with a range of individuals considering LDKT. Our goals in doing so were to consider general receptivity to *KidneyTIME*, and to evaluate potential differences in response to *KidneyTIME* among Black Americans, with an eye towards considering the potential to further develop *KidneyTIME* through including content specifically targeted for Black American viewers. We investigated patients' cognitive and communication barriers around LDKT and the potential of the videos to facilitate participant learning and sharing information, with specific emphasis on differences across race. Consideration of potential cultural differences is particularly important in LDKT educational research given long-standing concerns related to information accessibility that may influence the use of educational approaches.

We found that cognitive barriers of kidney failure patients to pursue LDKT were more prevalent among Black participants, primarily due to concern for the donor. Although concerns for donor health and well-being were a common barrier in both groups, such concerns were greater among Black participants, likely due to the well-known disproportionate burden

of kidney failure in this community [26] but perhaps also based on inaccurate beliefs and estimations [6]. Our findings of the similar incidence of other cognitive barriers are in contrast to prior research wherein Black kidney failure patients were more likely than other groups to downplay or outright deny the severity of their ESRD [7], to lack knowledge of LDKT benefits with respect to survival and quality of life [8], and to identify secondary benefits to remaining on dialysis [8]. These perspectives have been thought to serve an adaptive function but may delay or impede both access to treatment and taking steps toward receiving a living-donor kidney transplant. Although Black participants in our study more often expressed concern for donor health, concerns reduction appeared to align with new knowledge after viewing the videos. Previous research of health animation impact supports that the familiar and non-threatening nature of animation can facilitate the introduction of sensitive topics without inducing anxiety, thus enabling more comprehensive education [27–29].

We also found that communication barriers towards LDKT were reluctance and difficulty talking about LDKT. Both were more often described by Black than non-Black patients. Extensive studies have suggested that due to their own personal knowledge of risk factors and experience with kidney disease, Black kidney patients make a priori decisions about social network members' ineligibility to donate based on the potential donors' pre-existing medical conditions [9,10], family health history of inheritable diseases [10], and/or lifestyle [6,11], thereby precluding the recipient from initiating conversations [9] and accepting offers when they are made [6,11,12]. Other barriers to initiate discussions expressed by participants in our study and echoed by others include expectations of unsolicited offers to donate [13], fear of rejection of the request [12,13], discomfort approaching or asking [7,13], competency to approach [9], and concern of eliciting feelings of guilt or coercion from the social network [7]. Elements of the *KidneyTIME* animated video education that may support and facilitate communication about LDKT anticipated by both Black and non-Black participants were ability to share the videos asynchronously through social media or email, thereby increasing the capability of candidates to engage donors subtly, to avoid putting them on the spot. Both groups also described using the videos to start and inform conversations.

We also found anticipated usefulness of the videos to enhance general social support by educating social network members, reported by both race groups, as well as kidney failure patients, care partners, and donors. Previous studies have shown that prospective living donors may be challenged with barriers created by their own friends, family, and even intended recipients [23]. Prospective donors have reported having to defend themselves from friends and family, who persistently question their wisdom in donating, which can deter donation [23]. Similarly, patients need care partners and/or donor champions for decision-making and navigating the transplant process. Our findings suggest that the video explanations may make it easier to educate others to increase social support for donation and transplantation.

## Limitations

Our study has several limitations. The generalizability of our results may be limited to Black and predominantly non-Hispanic White adults, who have at least a high school education, live in Buffalo, NY, and who were referred to a transplant center. Differences that were found between race groups could also be a function of varying demographic factors that were different between Black and non-Black individuals, such as employment and income status, rather than race. Reporting was based on existing social support dynamics, which may have influenced our results, though most were individual interviews. The sample was made up predominantly of transplant recipients rather than those seeking a kidney transplant, which may have biased the identified concerns of transplant candidates. Although we drew data from all

participants, we only calculated proportion comparisons from patients to avoid an imbalanced race distribution of donors influencing results. Results may be confounded by obtaining participant perspectives about various numbers of videos in different stages of development and by the interviewer gender and race. As a secondary analysis, the interview questions may not have been optimal to investigate our study aims. Our interview approach was to facilitate participants' verbalization of their thoughts about a particular topic and probe from more context where a participant is leading. Therefore, we did not delve deeply into rationale around thoughts, feelings, and beliefs about a particular topic; this approach may have resulted in incomplete understanding. Nevertheless, to identify intervention adaptations, researchers traditionally conduct qualitative analyses after randomized trials or other systematic evaluations of effectiveness, which take considerable time to complete. A pragmatic solution to inform intervention modifications is to analyze qualitative data from development studies to obtain insights from the data that are specific to potential intervention adaptation [30]. Lastly, most of the interviewers lacked race concordance with Black participants, which may have impacted our data.

Limitations acknowledged, this secondary analysis of the original formative research and educational animated video development transcripts allowed the intervention designers to build a better understanding of the perspectives and beliefs of target users about LDKT and their views of the educational and potential outreach elements of the *KidneyTIME* intervention. Findings highlight the value of LDKT informational content that is appealing, digestible, and non-threatening, and is broadly shareable to both potential donors and other social network members through various media channels. Heterogeneity may also exist when considering access and intervention preferences in using the *KidneyTIME* animated educational videos, highlighting a potential for further cultural targeting or tailoring. Insights from the perspectives of the Black Americans suggest that intervention content should ensure that concerns for donor health are addressed, and may suggest a specific opportunity for culturally targeted adjunctive content. To further explore these findings and ensure future value, we will bring these concepts forward as a focus in our intervention refinement work.

## Acknowledgments

The authors would like to acknowledge the assistance of Danielle Wittek, Meriem Said, and Rachel Seibert during data collection.

## Statements & declarations

**Ethics approval.** This study was performed in line with the principles of the Declaration of Helsinki. Approval was granted by the Institutional Review Board of the University at Buffalo, State University of New York.

## Author Contributions

**Conceptualization:** Maria M. Keller, Thomas Feeley, Liise Kayler.

**Data curation:** Maria M. Keller, Liise Kayler.

**Formal analysis:** Maria M. Keller, Liise Kayler.

**Funding acquisition:** Liise Kayler.

**Investigation:** Maria M. Keller, Liise Kayler.

**Methodology:** Maria M. Keller, Liise Kayler.

**Project administration:** Maria M. Keller, Liise Kayler.

**Resources:** Liise Kayler.

**Software:** Maria M. Keller.

**Supervision:** Liise Kayler.

**Validation:** Thomas Feeley, Liise Kayler.

**Writing – original draft:** Maria M. Keller, Liise Kayler.

**Writing – review & editing:** Maria M. Keller, Todd Lucas, Renee Cadzow, Thomas Feeley, Laurene Tumiel Berhalter, Liise Kayler.

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
