## [Decision Letter · Decision Letter 0]

27 Jun 2022

PONE-D-22-09153Patient Perceptions by Race of Educational Animations About Living Kidney Donation Made for a Diverse PopulationPLOS ONE

Dear Dr. Keller,

Thank you for submitting your manuscript to PLOS ONE. After careful consideration, we feel that it has merit but does not fully meet PLOS ONE’s publication criteria as it currently stands. Therefore, we invite you to submit a revised version of the manuscript that addresses the points raised during the review process.

Please address the suggestions presented by the reviewer 2. Overall the reviewers were impressed with the manuscript, and I expect that the changes will not be too much of a burden.

We look forward to receiving your revised manuscript.

Kind regards,

Michael Scott Brewer, Ph.D.

Academic Editor

PLOS ONE

Journal Requirements:

Reviewers' comments:

Reviewer's Responses to Questions

**Comments to the Author**

1. Is the manuscript technically sound, and do the data support the conclusions?

Reviewer #1: Yes

Reviewer #2: Yes

2. Has the statistical analysis been performed appropriately and rigorously? 

Reviewer #1: Yes

Reviewer #2: Yes

3. Have the authors made all data underlying the findings in their manuscript fully available?

Reviewer #1: Yes

Reviewer #2: No

4. Is the manuscript presented in an intelligible fashion and written in standard English?

Reviewer #1: Yes

Reviewer #2: Yes

5. Review Comments to the Author

Reviewer #1: well written manuscript of a difficult topic. Look forward to followup with larger numbers.The analysis was quite detailed and robust. Very important for our field. Very important for out Black American patients.

Reviewer #2: Title

Abstract

Introduction

Very well written

I note the authors state that “Black Americans are disproportionately burdened by kidney failure, but are far less likely than any other racial or ethnic groups to undergo LDKT”

Are they classifying Black Americans as a separate race or ethnicity? If so, what do they mean by these terms? I note race is used hereafter

I note that the authors state “Black patients may need to reach out to a larger pool of potential living donors, since Black individuals who come forward as potential living donors are more likely to have disqualifying medical conditions”. Could the authors explain why Black people are only limited to kidneys from Black people?

I note that the transcripts being reviewed were already available when creating the first version of the videos. Could the authors explain why the original video creation may have excluded the views of Black participants, and why this re-review was needed?

Methods

Largely, well described

Do the authors believe their gender, race, or ethnicity affected the data collected? If so, how? Some sentences of reflection would be appreciated

Results

I note that Black patients had other demographic factors that were significantly different from non-Black patients. Could these other demographic factors be confounding results? Could the differences seen be a derivative of poverty and not race?

Did the authors explore why Black patients were specifically reluctant to talk about LDKT or found it difficult to talk about LDKT? Were there cultural aspects associated with this (i.e., ethnicity rather than race)? How do we tackle this issue?

Discussion

I note the authors make presumptions about the cognitive barriers present among Black patients – is there is a reason that the authors didn’t explore the rationale during the interview?

Overall, a good discussion of the results

6. PLOS authors have the option to publish the peer review history of their article (what does this mean?). If published, this will include your full peer review and any attached files.

Reviewer #1: No

Reviewer #2: No

---

## [Author Response · Author response to Decision Letter 0]

5 Aug 2022

Introduction

I note the authors state that “Black Americans are disproportionately burdened by kidney failure, but are far less likely than any other racial or ethnic groups to undergo LDKT”

Are they classifying Black Americans as a separate race or ethnicity? If so, what do they mean by these terms? I note race is used hereafter.

Response: Thank you for noting the need for clarification. We have indicated in the introduction that our term Black American encompasses individuals who self-identify as Black or as African American. This definition was reinforced in the methods section.

I note that the authors state “Black patients may need to reach out to a larger pool of potential living donors, since Black individuals who come forward as potential living donors are more likely to have disqualifying medical conditions”. Could the authors explain why Black people are only limited to kidneys from Black people?

Response: We did not mean to imply that Black people are limited to kidneys only from Black people and have revised the statement to better reflect the original intent that expanding the donor pool overall is needed.

I note that the transcripts being reviewed were already available when creating the first version of the videos. Could the authors explain why the original video creation may have excluded the views of Black participants, and why this re-review was needed?

Response: We have revised the introduction to more clearly indicate that we originally created a single educational product that was intended to affect most audiences; however, different message strategies and channel choices may be required to optimally influence different sub-groups. The decision on how to adapt should be made in the context of the degree of actual heterogeneity with regard to influences on behavior, channel choices, and responsiveness to different message executions.

Methods

Do the authors believe their gender, race, or ethnicity affected the data collected? If so, how? Some sentences of reflection would be appreciated

Response: Since all of the interviewers were female and all but one were non-Hispanic White, and the sole interviewer that was Black conducted a paucity of interviews, we were unable to investigate differences in data collection by interviewer demographic. We have indicated this limitation in the paper.

Results

I note that Black patients had other demographic factors that were significantly different from non-Black patients. Could these other demographic factors be confounding results? Could the differences seen be a derivative of poverty and not race?

Response: Thank you for illuminating this important point. We have updated our limitations section to indicate the following: Differences that were found between race groups could also be a function of varying demographic factors that were different between Black and non-Black individuals, such as employment and income status, rather than race.

Discussion

I note the authors make presumptions about the cognitive barriers present among Black patients – is there is a reason that the authors didn’t explore the rationale during the interview?

Response: We updated our methods to indicate that the interviews were semi-structured and that our analysis included probes related to the relevant questions. We have also added to the limitations section the possibility that we did not delve deeply into rationale around thoughts, feelings, and beliefs about a particular topic, possibly resulting in incomplete understanding.

Data Availability Statement is required.

Response: Our data availability statement is now provided in the manuscript. Limited data can be made available to researchers who meet the criteria for access to confidential data. Due to the qualitative nature of these data, the interview transcripts contain personal information that potentially identifies participants and would breach participant confidentiality if made publicly available. Data requests may be sent to the corresponding author. We will make our data available to researchers requesting the data directly from us. In that case, we will facilitate the submission of a short research proposal to the University at Buffalo, which also approved the original collection of the data. Once approved, we will happily share these data with future researchers for re-analysis. The University at Buffalo Internal Review Board can be reached at (716) 888-4888. 

By way of context, this study interviewed kidney failure patients and social network members at a single hospital. Release our data publicly is problematic since logical supposition of who the source is by members of the same community is possible and could jeopardize the anonymity of our participants and social network ties that arises from their shared history with others relating to kidney donation requests. In selecting particular quotes for the paper (grouped by theme rather than participant), we have ensured that no potentially identifiable information is included, but this information is woven throughout the interview transcripts and any suitably redacted or anonymized transcripts would be so unserviceably thin that they would be devoid of meaningful content.

---

## [Editor Report · Decision Letter 1]

24 Aug 2022

Patient Perceptions by Race of Educational Animations About Living Kidney Donation Made for a Diverse Population

PONE-D-22-09153R1

Dear Dr. Keller,

We’re pleased to inform you that your manuscript has been judged scientifically suitable for publication and will be formally accepted for publication once it meets all outstanding technical requirements.

Kind regards,

Michael Scott Brewer, Ph.D.

Academic Editor

PLOS ONE

Additional Editor Comments (optional):

Thank you for your patience throughout the review process. I had an abnormally difficult time finding qualified and willing reviewers. I think the manuscript was appreciably enhanced via the suggestions provided, so it was all worth it in the end.
---

## [Editor Report · Acceptance letter]

4 Sep 2022

PONE-D-22-09153R1 

Patient perceptions by race of educational animations about living kidney donation made for a diverse population 

Dear Dr. Keller:

I'm pleased to inform you that your manuscript has been deemed suitable for publication in PLOS ONE. Congratulations! Your manuscript is now with our production department. 

Kind regards, 

on behalf of

Dr. Michael Scott Brewer 

Academic Editor

PLOS ONE